# Dual-Stream STGCN with Motion-Aware Grouping for Rehabilitation Action Quality Assessment

**DOI:** 10.3390/s26010287

**Published:** 2026-01-02

**Authors:** Zhejun Kuang, Zhaotin Yin, Yuheng Yang, Jian Zhao, Lei Sun

**Affiliations:** 1College of Computer Science and Technology, Changchun University, Changchun 130022, China; kuangzhejun@ccu.edu.cn (Z.K.); 241501509@mails.ccu.edu.cn (Z.Y.); zhaojian@ccu.edu.cn (J.Z.); 2Jilin Provincial Key Laboratory of Human Health Status Identification Function & Enhancement, Changchun 130022, China; 3Key Laboratory of Intelligent Rehabilitation and Barrier-Free for the Disabled, Changchun University, Ministry of Education, Changchun 130022, China; 4College of Computer Science and Technology, Jilin University, Changchun 130012, China; maker220616@gmail.com

**Keywords:** artificial neural network, action quality assessment, physical rehabilitation

## Abstract

Action quality assessment automates the evaluation of human movement proficiency, which is vital for applications like sports training and rehabilitation, where objective feedback enhances patient outcomes. Action quality assessment processes motion capture data to generate quality scores for action execution. In rehabilitation exercises, joints typically work synergistically in functional groups. However, existing methods struggle to accurately model the collaborative relationships between joints. Fixed joint grouping is not flexible enough, while fully adaptive grouping lacks the guidance of prior knowledge. In this paper, based on rehabilitation theory in clinical medicine, we propose a dynamic, motion-aware grouping strategy. A two-stream architecture independently processes joint position and orientation information. Fused features are adaptively clustered into 6 functional groups by a joint motion energy-driven learnable mask generator, and intra-group temporal modeling and inter-group spatial projection are achieved through two-stage attention interaction. Our method achieves competitive results and obtains the best scores on most exercises of KIMORE, while remaining comparable on UI-PRMD. Experimental results using the KIMORE dataset show that the model outperforms current methods by reducing the mean absolute deviation by 26.5%. Ablation studies validate the necessity of dynamic grouping and the two-stream design. The core design principles of this study can be extended to fine-grained action-understanding tasks such as surgical operation assessment and motor skill quantification.

## 1. Introduction

Action quality assessment (AQA) is a task that evaluates the quality of action execution, often modeled as a fractional regression problem. Unlike tasks such as action recognition and action localization, AQA is a fine-grained action-understanding task that requires not only recognizing actions, but also distinguishing fine-grained differences between actions [1].

AQA is a computational task aimed at understanding and evaluating the quality of human movements [2], providing objective insights for various fields such as sports, rehabilitation, and skill training. In sports, AQA systems analyze video clips to rate performances in figure skating and diving [3,4,5]. In professional skill training, such as surgical training, AQA technology uses video assessments to provide detailed feedback on doctors’ movements, significantly improving the technical skills of residents [6,7]. The role of AQA in rehabilitation lies in its ability to provide quantitative metrics that reflect patient progress and the effectiveness of rehabilitation interventions [8,9]. Using AQA to analyze patients’ movements ensures they perform exercises correctly, minimizing the risk of further injury and maximizing recovery potential [10,11,12].

Early GCN approaches [13], while effective for action recognition, fail to capture time-varying functional connectivity in rehabilitation movements [14]. GCN can explicitly encode structural features of the skeleton by modeling joints and bones as nodes and edges of a graph, but its graph structure is usually fixed, making it difficult to capture high-order or dynamic spatial relationships between non-adjacent joints. Existing methods using convolutional neural networks do not make use of the spatial connectivity information of the human body, which limits the accuracy of these assessments [15].

Existing studies have improved evaluation accuracy through multimodal fusion [16,17], spatio-temporal attention mechanisms [18], and graph convolutional networks [14]. For instance, the Progressive Adaptive Multimodal Fusion Network proposed by Zeng et al. [16] leverages audio as a supplement to visual information, significantly enhancing the accuracy of score regression. Recent work by Chen et al. [17] combined inertial measurement units (IMUs) and surface electromyography (sEMG) with attention mechanisms to predict ligament fatigue, validating the efficacy of multimodal fusion in biomechanical assessment. This supports our dual-stream design for capturing complementary kinematic features. Meanwhile, Tube Self-Attention Network proposed by Wang et al. [18] generates rich spatio-temporal contextual information through sparse feature interactions, achieving optimal results on the Spearman’s rank correlation metric.

Precise AQA technology is crucial for health monitoring in an aging society. For instance, the KIMORE dataset [19] provides quantitative support for rehabilitation training by integrating kinematic data with clinical scores, but its real-time feedback capability is limited. In the field of sports, Liu et al. [20] evaluated figure skating jump movements by combining expert gaze positions with kinematic data, demonstrating the potential of AQA in improving training efficiency and competition fairness.

Current AQA research is mostly based on static skeleton topology or fixed joint grouping. Although it performs well in recognition and evaluation tasks, it is difficult to capture the dynamic changes in individual differences and joint synergy patterns of patients in the rehabilitation training process, resulting in a decline in accuracy on complex motion sequences. On the other hand, most existing methods only use position features, ignoring the important supplement of direction features to rehabilitation scoring, which limits the discrimination of the model. In rehabilitation, patients exhibit significant kinematic variability due to factors like injury severity or recovery stage. Fixed joint groupings fail to adapt to these individual differences, while ignoring orientation features (e.g., joint rotation angles) overlooks clinically critical aspects such as movement plane stability. For instance, in shoulder abduction exercises, scapulohumeral rhythm varies across patients, and rigid groupings cannot capture such dynamic synergies. This directly impacts assessment accuracy and personalized feedback.

The AQA method is typically evaluated using metrics such as Spearman’s rank correlation (ρ) to measure the alignment between predictions and ground truth values, Mean Squared Error (MSE) for regression tasks, and accuracy for classification methods. Performance is benchmarked on datasets such as UI-PRMD and KIMORE, where state-of-the-art methods achieve ρ = 0.80–0.90. However, current techniques face three key limitations: (1) Insufficient modeling of temporal dependencies in multi-phase movements, (2) poor generalization across different movement patterns, and (3) a reliance on handcrafted features, which reduces clinical interpretability. These gaps hinder deployment in real-world rehabilitation, where fine-grained feedback on movement phases is crucial for tracking patient progress. Our work addresses these issues in the following ways.

This paper proposes a dynamic functional grouping mechanism based on motion amplitude, which automatically aggregates joint groups with functional relevance through learnable masks. Combined with a two-stream architecture, it achieves complementary modality modeling, more closely aligning with the range of motion and stability of the plane of motion that physicians focus on in clinical rehabilitation assessments. To address the limitations of single-modality features, inflexible fixed grouping, and insufficient spatio-temporal modeling, we propose a three-stage framework: (1) A dual-stream architecture extracts and fuses joint position and orientation features to capture complementary motion semantics; (2) a motion-aware dynamic grouping mechanism replaces rigid topologies through learnable joint clustering; (3) two-stage attention interaction models intra-group dependencies and inter-group synergies. This design enables precise quantification of clinically significant rehabilitation patterns.

To summarize, the key contributions of our work are as follows:A feature extraction module that integrates joint position information and angle information is proposed.A dynamic grouping module is proposed to replace traditional fixed or uniform grouping. The adaptive allocation of joint nodes is implemented by a learnable mask generator, which differs from the fixed grouping of GCN. We group joint nodes based on the range of motion of each joint.A cross-head attention module is proposed to model intra-group spatial features and inter-group temporal features. Prior knowledge of motion magnitude is introduced into the model.Our method achieves the new state-of-the-art performance on two benchmark datasets and overall provides interesting insights. The implementations are released, hoping to facilitate future research.

## 2. Related Work

### 2.1. Graph Convolutional Network

Graph convolutional networks demonstrate significant advantages in action quality assessment due to the ability to model non-Euclidean data, such as human skeletal sequences. In the field of action quality assessment, a core and common research paradigm is dedicated to encoding hierarchical feature representations from skeleton sequences that can integrate spatial and temporal dimensions and take into account local and global information [21,22,23].

Pan et al. [24] proposed spatial relation graphs and temporal relation graphs to model the interactions of joints within a single frame and the coordination across time steps, respectively. They employed a Joint Commonality Module and a Joint Difference Module to learn the commonalities and differences in local joint movements. By using learnable adjacency matrices, the model’s interpretability was enhanced, leading to significant improvements in Spearman correlation coefficients for fine-grained action assessment tasks. However, the model relies on high-quality pose estimation results and has limited generalization capabilities in complex multi-person interaction scenarios.

Holmberg et al. [25] employed an Adaptive Adjacency Graph Convolutional Network (AAGCN) to analyze infant 3D skeletal sequences for predicting neurodevelopmental maturity, enhancing model flexibility through learnable adjacency matrices and attention mechanisms. The method outperforms traditional handcrafted feature-based approaches on 3D data and provides interpretable predictions. However, it has limited capability in modeling long-term temporal dependencies and does not account for dynamic scoring criteria of movement quality.

The foundation of skeleton-based rehabilitation assessment builds upon graph convolutional networks (GCNs). Seminal work by Yan et al. [13] introduced a Spatial–Temporal Graph Convolutional Network (STGCN) that modeled anatomical joint topology through fixed adjacency matrices. While effective for action recognition, these static graphs failed to capture dynamic functional connectivity patterns inherent in rehabilitation exercises [26]. Recent advances by Zhou et al. [14] addressed this limitation through learnable adjacency matrices, achieving 12.7% improvement on the KIMORE dataset [19]. Following previous work, we extracted spatial and temporal features from the original input. However, unlike the hierarchical model using STGCN mentioned above, we designed a motion magnitude-aware grouping mechanism and subsequently performed intra-group modeling and inter-group modeling through an attention mechanism.

### 2.2. Attention Mechanism

The application of attention mechanisms in AQA has gradually become mainstream, with its core advantage lying in the ability to capture long-term temporal dependencies and fine-grained spatio-temporal feature differences [27,28,29].

Traditional sliding window or segmentation methods struggle to model global temporal relationships in long videos, while attention mechanisms (such as the Transformer) expand the receptive field through self-attention layers, effectively addressing the fragmentation of cross-segment information. The literature [30] demonstrates the effectiveness of global attention mechanisms in 3D pose estimation, albeit at the cost of quadratic computational complexity. The literature [20,31] shows that self-attention-based temporal modeling improves the Spearman’s rank correlation score for diving action to 0.96.

TSA-Net [18] confines the self-attention mechanism within spatio-temporal tubes, enabling feature interaction only in local regions, significantly reducing computational complexity while preserving long-range dependencies. By combining tracking results with self-attention, it efficiently aggregates spatio-temporal context information, addressing issues like background noise interference and high computational costs in traditional methods. However, the model’s performance heavily relies on tracker accuracy—tracking failures may degrade feature aggregation effectiveness.

Unlike the strategy of spatio-temporal aggregation only in local regions as in reference [18], we use a local–global hybrid mechanism to model global temporal features while ensuring local spatial modeling. Different from the temporal attention aggregation strategy proposed in reference [32], we introduce interpretable constraints based on motion amplitude in cross-modal data (position information and direction information) to increase the physical rationality of attention weights.

## 3. Method

### 3.1. Problem

Equation (Equation 1) represents a sequence of skeletal joint data from a rehabilitation exercise, characterized by a duration of *T* and *J* collected joints. The vector $x_{j,t}^{pos}$ is a three-dimensional vector [x,y,z] specifying the position coordinates of joint *j* at time *t* within a Cartesian coordinate system. Concurrently, the quaternion $x_{j,t}^{ori}=\left[w,u^{x},u^{y},u^{z}\right]$ represents the orientation of joint *j* at time *t* as a rotation, where *w* is the cosine of half the rotation angle and $\left[u^{x},u^{y},u^{z}\right]$ forms a unit vector delineating the rotation axis (see Figure 1). (1)$$X=\left\{X^{pos},X^{ori}\right\}={\left\{x_{j,t}^{pos},x_{j,t}^{ori}\right\}}_{j=1,t=1}^{J,T}.$$

The goal of the task is to return a sequence of successive actions to a score *Y*, which can be mathematically defined as(2)Y=f(X;θ),
where f(·) represents the neural network model, and θ denotes the parameters of the model. During training, Equation (Equation 3) is used to measure the error between the model’s predicted output and the ground truth. Backpropagation is employed to compute gradients, and gradient descent is applied to update the parameters θ, allowing them to gradually converge to a local optimum. The mathematical definition of this loss function is as follows:(3)$$L\left(\theta \right)=\frac{1}{N}\sum_{i=1}^{N}\ell \left(f\left(x_{i};\theta \right),y_{i}\right),$$
where *N* is the batch size, $\ell \left(\cdot \right)=\frac{1}{2}{∥\hat{y}-y∥}_{2}^{2}$ is the sample-level loss.

### 3.2. Dual-Stream STGCN Network

The proposed Dual-Stream STGCN architecture aims to capture spatio-temporal features from joint positions and orientations. The dual-stream architecture processes positional coordinates $X^{pos}\in R^{T\times J\times 3}$ and orientation quaternions $X^{ori}\in R^{T\times J\times 4}$ through parallel STGCN branches. A single STGCN module follows a three-stage hierarchy consisting of spatial graph convolutional, temporal convolution, and residual connection.

In the first stage of model training, the model aggregates features in the spatial domain of the action sequence while capturing temporal characteristics. The hierarchical architecture uses three cascaded STGCN blocks to expand the channel count from 64 to 256 for both branches. The last two STGCN blocks perform downsampling with a stride of 2 in the temporal dimension. The positional and directional information from the two branches are concatenated along the channel dimension.

Spatial aggregation is achieved by lightweight 2D convolution (kernel size is set to 1 times 1) operating along the spatial dimension, using convolution kernels to perform feature transformations along the joint dimension, which is equivalent to performing a fully connected transformation on each joint individually [13].

Temporal convolution uses 2D convolution along the time dimension, with a kernel size of 3 times 1 to capture a 3-frame time window. A stride of 2 is used for downsampling in the time dimension, and padding of 1 is used to maintain feature map alignment.

A temporal kernel $K_{t}\in R^{3\times C_{in}\times C_{out}}$ captures local features:(4)$$F_{temp}=\mathrm{Conv}2D\left(F_{spatial},K_{t}\right)\in R^{T/2\times J\times C_{out}}.$$

Residual connection adjusts feature variance, suppressing gradient vanishing:(5)$$X_{out}=\mathrm{ReLU}\left(X_{temporal}+X_{residual}\right).$$

Position and orientation stream outputs are concatenated along the channel dimension:(6)$$X=\mathrm{Concat}\left(X^{\mathrm{pos}},X^{\mathrm{ori}}\right)\in R^{B\times T^{\prime }\times J\times C},\;T^{\prime }=\left\lceil \frac{T}{4}\right\rceil ,$$
where $X_{pos}\in R^{B\times T\times J\times 256}$ and $X_{ori}\in R^{B\times T\times J\times 256}$ denote feature streams that encode positional and directional information, respectively. These streams are concatenated along the channel dimension to form the fused feature representation *X*.

### 3.3. Self Attention for Dynamic Groups

The proposed Self-Attention for Dynamic Groups (SADG) module aims to model spatio-temporal features from joints and groups. In rehabilitation training, joints typically work synergistically in functional groups. However, traditional fixed grouping cannot adapt to individual patient differences. In our current implementation, the range of motion index serves as a computationally efficient proxy metric to prioritize joints with higher kinematic variability. Our dynamic grouping strategy automatically identifies functionally related joint clusters through the range of motion, thereby more accurately capturing the synergistic patterns during the rehabilitation process. We quantify the dynamic variation of each joint point for specific movements by analyzing the positional changes in joint points along the time dimension, where *M* indicates the range of the motion:(7)$$M=\sqrt{\frac{1}{T}\sum_{t=1}^{T}{\left(X_{b,t,j,c}-\mu_{b,j,c}\right)}^{2}}\in R^{B\times J\times C},$$
$X\in R^{B\times T\times J\times C}$ is the input feature tensor, through which the motion intensity is captured by calculating the standard deviation along the time axis *T*, where $\mu_{b,j,c}=\frac{1}{T}\sum_{t=1}^{T}X_{b,t,j,c}$ represents the mean along the time dimension. As shown in Table 1, we list the assignment probabilities for 25 joints corresponding to 6 groups. It should be noted that the joint assignment here is a soft assignment.

The dynamic group mask α is generated through a two-stage linear transformation that maps the range of joint motion *M* to group assignment probabilities:(8)$$\alpha ={\mathrm{Softmax}}_{g}\left(W_{2}\cdot \mathrm{ReLU}\left(W_{1}\cdot M+b_{1}\right)+b_{2}\right),$$

The learnable parameters $\left\{W_{1},W_{2}\right\}$ implement feature projection across hidden dimensions, where *d* denotes the latent representation space dimensionality. The bias terms $\left\{b_{1},b_{2}\right\}$ introduce translational invariance to accommodate inter-subject kinematic variations. The resulting tensor α represents probabilistic assignments of *J* anatomical joints to *G* functional groups. We conducted ablation study on the impact of the number of groups on model performance (table in Section 6), and from this, we determined the number of groups to be 6.

Intra-group attention captures joints activation patterns while inter-group projection models motions across functional groups.(9)$$X_{b,t,g,c}^{\left(g\right)}=\sum_{j=1}^{J}\alpha_{b,g,j}X_{b,t,j,c},$$
the formula is actually to project the joint *J* dimension to the group *G* dimension. The purpose of this operation is to aggregate the features of each joint into the corresponding group according to the group-assigned probability weighting.(10)$$A_{b,t,g,h,k}={\mathrm{Softmax}}_{k}\;\left(\frac{1}{\sqrt{C}}\sum_{d=1}^{D}Q_{b,t,g,h,d}\;K_{b,t,g,k,d}\right),$$
where $Q\in R^{B\times T\times G\times H\times D}$ is query matrix, obtained by grouping feature $X_{\mathrm{group}}$ via linear transformation. $K\in R^{B\times T\times G\times H\times D}$ is the key matrix. *D* is the dimension of key vector, D=C/H.

We concatenate the H head outputs along the head dimension to obtain the group feature, where $V\in R^{B\times T\times G\times K\times D}$ is matrix of values for the dynamic representation of joint features:(11)$$Z_{b,t,g,h,d}=\sum_{k=1}^{K}A_{b,t,g,h,k}\;V_{b,t,g,k,d}.$$

The fused feature maintains consistency with the number of input channels and provides conditions for residual connection. The fused group representation is redistributed to the original joint through a learnable mask, retaining the “local coordination–global compensation” principle [19]. The grouping feature is ${\tilde{Z}}^{\left(g\right)}\in R^{B\times T\times G\times C}$ and the grouping mask is $\alpha \in R^{B\times G\times J}$, then the back projection operation is(12)$$Z_{b,t,j,c}=\sum_{g=1}^{G}{\tilde{Z}}_{b,t,g,c}^{\left(g\right)}\cdot \alpha_{b,g,j}\in R^{B\times T\times J\times C}.$$

The grouped features are mapped back to the original joint dimensions to preserve the tensor structure, which ensures that subsequent layers can directly process standard skeletal structures. As shown in Figure 2, we visualized the attention weights at different time steps. As shown in Figure 3, we also visualized the attention weights for different exercises on KIMORE.

### 3.4. Regression Head

The regression head is a sequence of two linear layers, ReLU activation. The input dimension is set to 512; after the first linear layer is set to 128, then ReLU, the output dimension, is 1 because it is a regression task. The linear layer is responsible for transforming the feature space, while ReLU introduces nonlinearity:(13)$$f_{reg}\left(x\right)=W_{2}\cdot \left(\sigma \left(W_{1}\cdot x+b_{1}\right)\right)+b_{2},$$
where $x\in R^{512}$ is the input feature, $W_{1}\in R^{128\times 512}$, and $b_{1}\in R^{128}$, σ is the ReLU activation function, $W_{2}\in R^{1\times 128}$, $b_{2}\in R$. The first linear layer projects high-dimensional spatio-temporal features (512D) into lower dimensions (128D), which suppresses noise by dimensionality reduction and improves generalization. The second linear layer outputs the final score. The dimensional compression ratio $\frac{128}{512}=0.25$. Experience shows that this ratio strikes a balance between information retention and overfitting control.

## 4. Experiments

We adopt three complementary metrics:

MAD (Mean Absolute Deviation): By quantifying the average deviation from actual action quality scores, it provides a simple and intuitive measure of prediction accuracy:(14)$$MAD=\frac{1}{N}\sum_{i=1}^{N}\left|y_{i}-{\hat{y}}_{i}\right|,$$
where $y_{i}$ are the actual values, $\hat{y_{i}}$ are the predicted values, and *N* is the number of observations.

RMSE (Root Mean Square Error): It is particularly useful when large errors are highly undesirable. It quantifies the square root of the average squared difference between predicted and actual values:(15)$$RMSE=\sqrt{\frac{1}{N}\sum_{i=1}^{N}{\left(y_{i}-{\hat{y}}_{i}\right)}^{2}}.$$

MAPE (Mean Absolute Percentage Error): Provides a scale-independent measure of accuracy, facilitating comparisons across different datasets or models:(16)$$MAPE=\frac{100\%}{N}\sum_{i=1}^{N}\left|\frac{y_{i}-{\hat{y}}_{i}}{y_{i}}\right|.$$

SRC (Spearman’s Rank Correlation): The Spearman correlation indicates that the model accurately predicted the order of actions based on quality scores, which is crucial for ranking and prioritization tasks. It calculates the correlation based on the rank values of the variables rather than their original values:(17)$$SRC=\frac{\sum_{i=1}^{n}\;\left(Q_{i}-\bar{Q}\right)\left(Y_{i}-\bar{Y}\right)}{\sqrt{\left(\sum_{i=1}^{n}\;{\left(Q_{i}-\bar{Q}\right)}^{2}\right)\left(\sum_{i=1}^{n}\;{\left(Y_{i}-\bar{Y}\right)}^{2}\right)}},$$
where *Q* and *Y* are the mean ranks of the predicted and actual values, respectively.

### 4.1. Experimental Settings

We used the KIMORE dataset [19] and the UI-PRMD dataset [33] for training and evaluation, and the KIMORE dataset for ablation experiments. Experiments on the KIMORE dataset used a fixed random seed [34], while experiments on the UI-PRMD dataset did not fix the random seed and averaged the results over five runs.

The KIMORE rehabilitation assessment dataset [19] contains 3827 sample sequences across five clinically significant exercises. The first (ex1) involves upper-limb extension movements, the second (ex2) involves lateral flexion movements of the upper limbs and trunk, the third (ex3) involves rotational movements of the upper limbs and trunk, the fourth (ex4) involves rotational movements of the pelvis in the transverse plane, and the last one (ex5) involves squat movements of the lower limbs and trunk. KIMORE uses the Microsoft Kinect v2 as its RGB-D visual sensor. Subjects were required to continuously repeat each exercise 5 times. They were placed 3 m in front of the Kinect sensor, and distances and angles were calculated in the frontal and sagittal planes, respectively. The Microsoft Windows v2 sensor uses Time-of-Flight (ToF) technology, whereas the previous sensor (Kinect v1) belongs to the category of structured light (SL) cameras. Compared to cameras based on SL technology, ToF cameras have a longer range, and the image appears more accurate when there are no holes in the depth map. Compared to the previous version, the Kinect v2 provides a higher depth map resolution (512 × 424 vs. 320 × 240), enabling it to recognize thin objects and resolve some ambiguities. Depth features allow for the recognition of different subjects and body parts within the field of view, while the increased resolution allows for the recognition of 3D points of 25 different body parts at 30 fps. Each sample provides synchronized 3D positional coordinates and orientation quaternions for 25 joints, captured at 30 Hz with sequence lengths normalized to 100 frames through cubic interpolation. The KIMORE label score (0 to 100) is obtained by clinicians and experts in musculoskeletal and neurological diseases evaluating each exercise (Cohen’s Kappa test which reached a K-value > 0.8 [35]). To ensure class balance, we implement stratified sampling, with samples divided 8:1:1 into train/validation/test sets.

The UI-PRMD dataset [33] involves 10 healthy subjects performing 10 repetitions of different physical therapy movements using a Vicon optical tracker and a Microsoft Kinect sensor for motion capture. The label score (0 to 100) of UI-PRMD is a mean squared deviation value that has undergone time and dimension normalization. As an objective quantitative metric, it is primarily used to study the consistency and variability of movement execution in physical rehabilitation exercises, providing benchmark data for assessing patient rehabilitation progress or algorithm performance. As shown in Table 2, we list the exercise names and movement descriptions. The movements in UI-PRMD were obtained simultaneously through Vicon and Kinect systems. The software programs Nexus 2 and Brekel recorded the actions using Vicon and Kinect systems, respectively. The motion capture frame rate for Vicon is 100 Hz, while the frame rate for Kinect is 30 Hz. For measurements from both Vicon and Kinect, the Cartesian position values of the joints are in millimeters, and the joint angles are in degrees. The values in the dataset are presented in record form. The only preprocessing step was to correct large jumps in the Kinect angle measurements, because the angles are limited to the range (−180°, +180°). For values exceeding these limits, the value continues on the other side of the limit. No other data processing was performed. The Vicon optical tracker captures 39 human joints, while the Kinect sensor captures 22. We conducted experiments using the data captured by the Vicon optical tracker. Unlike the KIMORE dataset, the joint angles in UI-PRMD are quantified using Euler angles, which, like the joint position information, are represented as triplets. The dataset division follows the same stratified sampling method as KIMORE. On UI-PRMD, we did not use normalization.

### 4.2. Implementation Details

We conduct experiments on a computer equipped with an Intel i5 CPU at 2.5 GHz (Santa Clara, CA, USA), NVIDIA GeForce GTX 5060 GPU (Santa Clara, CA, USA), and the RAM of 32 GB. We use PyTorch 2.8 and CUDA 12.9 to implement our model. We apply preprocessing method of z-score to the data in KIMORE.

We use grid search to perform hyperparameter tuning on the two datasets separately. Both datasets use a cosine annealing learning rate scheduler, with KIMORE using AdamW as the optimizer and a weight decay of 0.0001, and UI-PRMD using SGD as the optimizer without weight decay. For the KIMORE dataset, the initial learning rate is set to 0.0001, decaying by 0.0001 per epoch, and the batch size is set to 32. For the UI-PRMD dataset, the initial learning rate is set to 0.1, and the batch size is set to 1. The two datasets are completely separate, and training and evaluation are also conducted independently for each dataset. We trained for 100 epochs on each of the two datasets, respectively. As shown in Figure 4, we visualized the training loss of the model on UI-PRMD’s ex1 after 100 epochs.

## 5. Comparison with Existing Methods

Following evaluation metrics from previous work, our model is trained on each of the five exercises in KIMORE dataset. In Table 3, we show the performance comparison of our proposed method with other state-of-the-art methods on the KIMORE dataset. Our model, which combines multimodal spatio-temporal feature extraction for positional and orientational information, a dynamic grouping attention mechanism based on motion amplitude, outperforms competing methods in most exercises.

As evidenced in Table 3, D2STA achieves superior performance across most rehabilitation exercises in the KIMORE dataset, with an average MAD of 0.119 representing a 26.5% improvement over the best existing methods. This performance enhancement primarily stems from the synergistic interaction between the dynamic functional grouping strategy and dual-stream architecture.

Specifically, our method achieved the lowest MAD, RMSE, and MAPE across most exercises in the KIMORE dataset. Compared to the state-of-the-art methods, our model generally performed better. Only on ex4 did our model exhibit slightly higher error than the model by Kuang et al. [9]. Our model’s performance on ex4 is relatively poorer compared to other actions, which may depend on the ex4 action itself. ex4 is a rotation action of the pelvis in the transverse plane. Compared to other actions, the range of motion for ex4 is relatively small, and the model may not be able to accurately capture the action features. Overall, our model demonstrated significant improvements, reducing MAD by approximately 26.52%, RMSE by 16.76%, and MAPE by 33.86% compared to the best existing methods. These results highlight the accuracy and effectiveness of our approach in action quality assessment.

To further demonstrate the robustness of our model, we evaluated it on the UI-PRMD dataset. UI-PRMD provides positional (39 joints) and orientational (22 joints) data with incompatible dimensionality. To ensure input consistency, only orientation features were used. We maintained the same model architecture, with all inputs being directional data. For the UI-PRMD, the data was not preprocessed in any other way, ensuring consistency with the methods of others, such as [31,36]. As shown in Table 4 and Table 5, our model achieves competitive results compared to the existing best methods. These results demonstrate the effectiveness and robustness of our model, even when directly applied to other similar datasets without data normalization and using only unimodal data.

As shown in Figure 5, we visualize the correlation between the model’s predicted scores and the actual scores on the ex1 of KIMORE. The strong agreement between the predicted and actual values is evident, demonstrating the model’s accuracy. The Spearman’s Rank Correlation further quantifies this relationship, yielding a high correlation coefficient of 0.9497. This high SRC value indicates a strong correlation between the predicted and actual scores. The Spearman’s Rank Correlation for the other exercises, ex2 to ex5, are 0.9643, 0.9849, 0.9635, and 0.9631, respectively. This high SRC value indicates that the model consistently assigns high scores to high-quality performances and low scores to low-quality performances.

## 6. Ablation Study

Ablation experiments use the same splits. To ensure reproducibility and account for the impact of random seeds on model performance, we fix the random seed to 3407 [34] and unify the training parameters: epoch is set to 100, learning rate is set to 0.0001, batch size is set to 32.

D2STA is regarded as the baseline model, aiming to evaluate its performance without the SADG module. Additionally, we replace the SADG module with an attention mechanism lacking dynamic grouping to verify the effectiveness of the dynamic grouping module. The effectiveness of the dual-stream architecture is verified by changing the dual-stream STGCN to single-stream STGCN.

As shown in Table 6, the experiments were conducted on exercises ex1 to ex5 of the KIMORE dataset, with the results averaged across the five exercises. The experimental results indicate that the absence of SADG leads to increased errors in MAE, RMSE, and MAPE. Replacing the SADG module with multi-head self-attention without dynamic grouping further degrades the performance of these metrics.

When replacing the dual-stream STGCN with a single-stream STGCN, the performance significantly decreases. The single-stream architecture resulted in a 64% increase in MAPE, demonstrating the irreducible complementarity of the dual modality information. Ablation studies in Table 6 reveal that removing the dynamic grouping module increases MAPE by 90%. The dual-stream architecture’s necessity is validated by ablation experiments showing RMSE increases from 0.221 to 0.406 when using single-stream processing (Table 6). This degradation occurs because clinical scoring of rehabilitation movements simultaneously depends on joint trajectory (position stream) and movement plane stability (orientation stream). Through multimodal cascading, the dual-flow design makes up for the lack of single-modal information.

As shown in Table 7, we conducted an ablation study on the number of groups for the dynamic grouping module. With other parameters held constant, the grid search method was used to determine the number of groups to be 6 by only changing the group count.

As shown in Table 8, we conducted an ablation study on the number of attention heads. With a constant total number of channels, too few attention heads may fail to capture multi-scale features, while too many may lead to insufficient representational capacity for each head.

## 7. Conclusions

This study proposed a dual-stream spatio-temporal attention-based rehabilitation action quality assessment framework. By integrating dynamic functional grouping with spatio-temporal modeling, the proposed framework achieves competitive performance on the KIMORE dataset and the UI-PRMD dataset. The dual-stream independent modeling of position and orientation preserves modality specificity by soft-assigning joint points into 6 groups and then modeling intra-group spatial features and inter-group temporal features, thereby supplementing traditional spatio-temporal modeling methods. The ablation experiment shows that the single-flow architecture leads to a 90% increase in MAPE, verifying the necessity of the dual-stream design. Removing the SADG module caused a significant increase in MAPE, validating the effectiveness of the SADG module design. These results have provided intriguing insights overall and are hoped to promote the development of the community.

The dynamic functional grouping mechanism based on the range of motion automatically aggregates joint groups with functional correlations through learnable masks, more closely aligning with the range of motion and the stability of the motion plane in rehabilitation exercises. It is worth noting that, although the motion range index in this paper can effectively account for highly variable joints, it does not explicitly model the physiological constraints of specific joints (e.g., natural range of motion), nor does it normalize the displacement based on the body’s topological structure. Future work will integrate normalization relative to the root node and a clinical ROM database to address these limitations, especially for low-amplitude but high-precision joints that are crucial in cervical spine or hand rehabilitation.

## Figures and Tables

**Figure 1 sensors-26-00287-f001:**
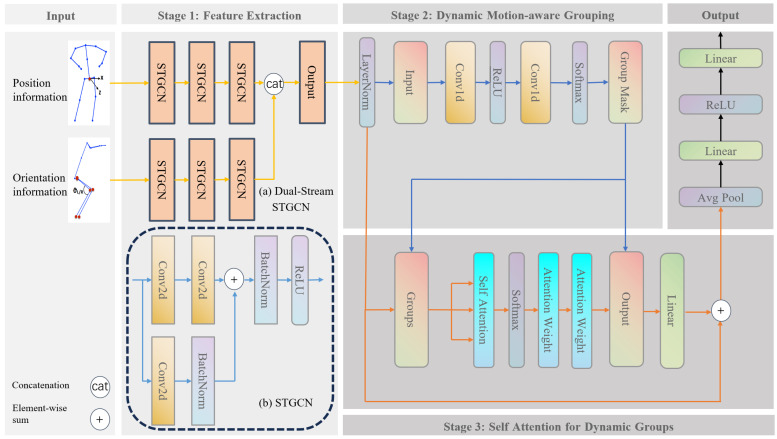
The overall pipeline of our network. The goal is to predict a continuous quality score from a sequence of human actions. Initially, we extract spatio-temporal features by concatenating positional and directional information along the channel dimension, followed by applying a group-wise mask through matrix multiplication. Subsequently, these grouped features are processed through a two-layer attention mechanism for effective feature fusion, though only one attention module is illustrated for clarity. Finally, the attention weights are projected back to the original skeleton structure via matrix multiplication with the group mask. After establishing a residual connection with the initial features, the resulting representation is passed to a regression head to generate the final quality score.

**Figure 2 sensors-26-00287-f002:**
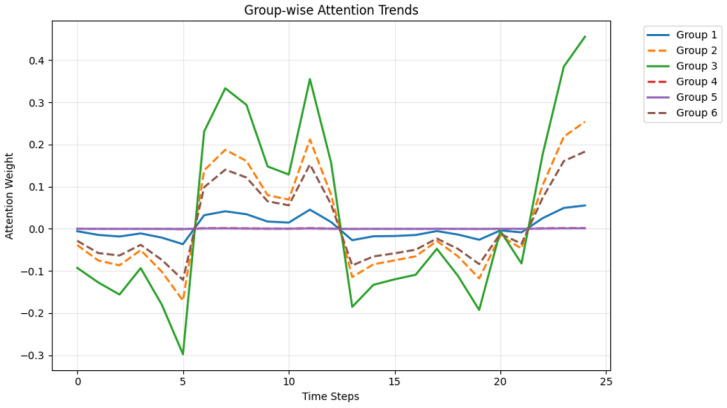
Different groups have different attention weights at different time steps.

**Figure 3 sensors-26-00287-f003:**
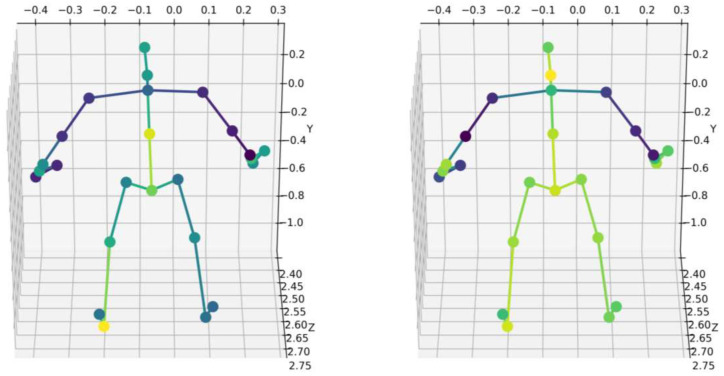
The left figure shows the attention weights visualization on ex2, and the right figure shows the attention weights visualization on ex5.

**Figure 4 sensors-26-00287-f004:**
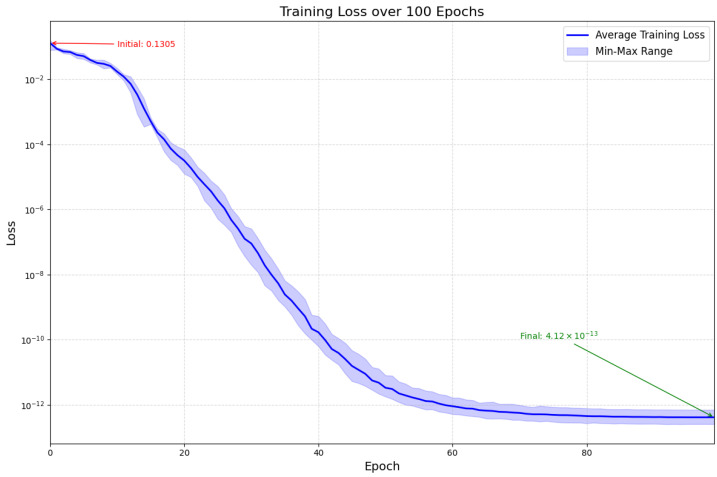
The training loss for the model over 100 epochs, where the dark curve in the figure represents the average loss value from 3 experiments, and the light-colored area’s upper and lower bounds are the maximum and minimum values, respectively.

**Figure 5 sensors-26-00287-f005:**
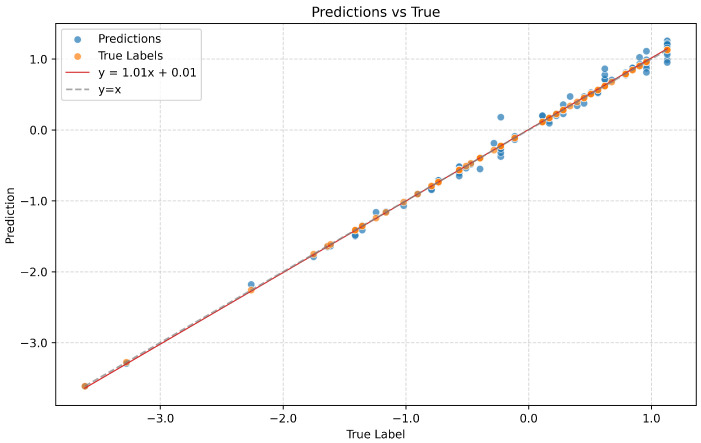
The correlation between the model’s predicted scores and the actual scores on the ex1 of KIMORE.

**Table 1 sensors-26-00287-t001:** The group assignment probability for the joints of ex1 on KIMORE.

Joint	Group 1	Group 2	Group 3	Group 4	Group 5	Group 6
Spine_Base	0.155	0.148	0.172	0.174	0.172	0.178
Spine_Mid	0.153	0.156	0.161	0.184	0.183	0.163
Neck	0.156	0.148	0.146	0.191	0.200	0.159
Head	0.154	0.158	0.151	0.187	0.187	0.164
Shoulder_L	0.147	0.164	0.158	0.192	0.167	0.171
Elbow_L	0.151	0.149	0.159	0.174	0.183	0.183
Wrist_L	0.149	0.144	0.157	0.185	0.204	0.162
Hand_L	0.144	0.153	0.151	0.189	0.196	0.166
Shoulder_R	0.138	0.152	0.159	0.201	0.183	0.167
Elbow_R	0.150	0.156	0.157	0.198	0.180	0.158
Wrist_R	0.152	0.161	0.163	0.187	0.178	0.159
Hand_R	0.157	0.157	0.171	0.187	0.178	0.159
Hand_R	0.157	0.157	0.165	0.180	0.171	0.171
Hip_L	0.157	0.150	0.162	0.186	0.173	0.172
Knee_L	0.153	0.156	0.156	0.185	0.173	0.177
Ankle_L	0.160	0.155	0.164	0.181	0.170	0.171
Foot_L	0.158	0.150	0.158	0.183	0.184	0.166
Hip_R	0.156	0.146	0.153	0.207	0.188	0.150
Knee_R	0.153	0.161	0.148	0.191	0.180	0.167
Ankle_R	0.154	0.160	0.154	0.182	0.177	0.173
Foot_R	0.156	0.152	0.159	0.183	0.177	0.173
Spine_Shoulder	0.145	0.147	0.157	0.191	0.183	0.182
Thumb_L	0.141	0.149	0.148	0.194	0.195	0.168
Tip_L	0.145	0.150	0.147	0.202	0.199	0.152
Thumb_R	0.149	0.159	0.159	0.192	0.176	0.148

**Table 2 sensors-26-00287-t002:** Exercises and descriptions in the UI-PRMD dataset.

Exercise	Movement	Description
ex1	Deep squat	Subject bends knees to descend body toward floor with heels on floor, knees aligned over feet, upper body remains vertical
ex2	Hurdle step	Subject steps over hurdle while hips, knees, and ankles of standing leg remain vertical
ex3	Inline lunge	Subject takes step forward and lowers body toward floor to make contact with knee behind front foot
ex4	Side lunge	Subject takes step to side and lowers body toward floor
ex5	Sit to stand	Subject lifts body from chair to standing position
ex6	Standing active straight leg raise	Subject raises one leg in front of body while keeping leg straight and body vertical
ex7	Standing shoulder abduction	Subject raises one arm to side by lateral rotation, keeping elbow and wrist straight
ex8	Standing shoulder extension	Subject extends one arm rearward, keeping elbow and wrist straight
ex9	Standing shoulder internal–external rotation	Subject bends elbow to 90° angle, and rotates forearm forward and backward
ex10	Standing shoulder scaption	Subject raises one arm in front of chest to shoulder height, keeping elbow and wrist straight

**Table 3 sensors-26-00287-t003:** Performance comparison on KIMORE dataset: the best values being bolded. ‘-’ indicates the vacancy value not reported.

Methods	MAD					RMSE					MAPE				
	ex1	ex2	ex3	ex4	ex5	ex1	ex2	ex3	ex4	ex5	ex1	ex2	ex3	ex4	ex5
Du et al. (2021) [15]	1.271	2.199	1.123	0.880	1.864	2.440	4.297	1.925	1.676	3.158	3.228	6.001	3.421	2.584	5.620
Deb et al. (2022) [36]	0.799	0.774	0.374	0.347	0.621	2.024	2.120	0.556	0.644	1.181	1.926	1.272	0.728	0.824	1.591
Yao et al. (2023) [37]	0.444	0.303	0.142	0.121	0.292	0.569	0.390	**0.180**	0.148	0.378	1.105	0.864	0.437	0.341	0.808
Mour et al. (2023) [31]	0.641	0.753	0.210	0.206	0.399	2.020	1.468	0.487	0.527	0.735	1.623	0.974	0.613	0.541	1.217
Sardari et al. (2024) [11]	0.200	0.270	0.210	0.280	0.250	0.250	0.320	0.190	0.300	0.270	-	-	-	-	-
Xiao et al. (2024) [38]	-	-	-	-	0.429	-	-	-	-	0.953	-	-	-	-	1.130
Zhang et al. (2025) [39]	0.622	0.491	0.206	0.204	0.390	1.387	0.748	0.398	0.515	0.698	1.508	0.952	0.536	0.483	1.113
Kuang et al. (2025) [9]	0.186	0.235	0.111	**0.053**	0.223	0.399	0.354	0.289	**0.101**	0.386	0.431	0.749	0.271	**0.173**	0.692
Ours	**0.102**	**0.119**	**0.100**	0.163	**0.110**	**0.162**	**0.194**	0.407	0.148	**0.196**	**0.187**	**0.297**	**0.207**	0.257	**0.585**

**Table 4 sensors-26-00287-t004:** Performance comparison of different methods on UI-PRMD dataset based on MAD: the best values being bolded.

Methods	MAD									
	ex1	ex2	ex3	ex4	ex5	ex6	ex7	ex8	ex9	ex10
Deb et al. (2022) [36]	0.009	**0.006**	0.013	**0.006**	0.008	0.006	0.011	0.016	0.008	0.031
Mour et al. (2023) [31]	0.011	0.009	0.013	0.009	0.009	0.013	0.022	0.020	0.013	0.014
Sardari et al. (2024) [11]	0.014	0.007	0.011	**0.006**	0.008	0.006	**0.010**	**0.011**	0.008	0.038
Kuang et al. (2025) [9]	0.010	**0.006**	**0.008**	0.008	**0.007**	**0.005**	0.012	**0.011**	**0.006**	0.018
Ours	**0.008**	0.010	0.015	**0.006**	0.010	0.007	**0.010**	0.014	0.012	**0.013**

**Table 5 sensors-26-00287-t005:** Performance comparison of different methods on UI-PRMD dataset based on RMSE and MAPE: the best values being bolded.

Metrics	Methods	ex1	ex2	ex3	ex4	ex5	ex6	ex7	ex8	ex9	ex10
RMSE	Deb et al. (2022) [36]	0.020	0.016	0.024	0.015	0.014	0.025	0.036	0.034	0.022	0.033
	Mour et al. (2023) [31]	0.019	0.014	0.020	0.011	0.013	0.020	0.034	0.032	0.019	0.023
	Kuang et al. (2025) [9]	0.016	**0.008**	**0.013**	0.011	**0.009**	**0.007**	0.019	**0.015**	**0.009**	0.031
	Ours	**0.009**	0.015	0.018	**0.009**	0.012	0.008	**0.013**	0.018	0.015	**0.016**
MAPE	Deb et al. (2022) [36]	1.337	1.244	1.758	1.090	1.176	1.994	2.980	2.815	1.873	2.900
	Mour et al. (2023) [31]	1.289	1.105	1.592	0.984	1.032	1.476	2.697	2.362	1.455	1.619
	Kuang et al. (2025) [9]	1.161	**0.659**	**0.925**	0.875	**0.734**	**0.513**	1.356	**1.263**	**0.670**	2.172
	Ours	**0.811**	1.361	1.560	**0.794**	1.040	0.746	**1.163**	1.566	1.038	**1.365**

**Table 6 sensors-26-00287-t006:** Ablation Study: The baseline model is D2STA. Position stream represents the model with the orientation branch removed, and orientation stream represents the model with the orientation branch removed. MHSA represents the model with the grouping module removed, and Without SADG represents the model with the SADG module removed. The best value is bold. A check mark indicates that position data (pos) or orientation data (ori) is used, while a cross mark indicates the opposite.

	pos	ori	MAD	RMSE	MAPE
Baseline	✓	✓	**0.1187**	**0.2214**	**0.3064**
Position stream	✓	✗	0.2308	0.4064	0.8446
Orientation stream	✗	✓	0.2851	0.5342	0.9501
MHSA	✓	✓	0.1776	0.2742	1.0746
Without SADG	✓	✓	0.2464	0.3752	2.9218

**Table 7 sensors-26-00287-t007:** Ablation study: The impact of the number of groups in the dynamic grouping module on performance.

Groups	2	4	6	8	10
MAD	0.548	0.179	0.119	0.208	0.741
RMSE	0.379	0.228	0.194	0.254	0.502
MAPE	1.523	0.285	0.297	0.439	1.826

**Table 8 sensors-26-00287-t008:** Ablation study: The impact of the number of heads in the SADG module on performance.

Heads	4	6	8	10	12
MAD	0.399	0.156	0.119	0.211	0.317
RMSE	0.372	0.187	0.194	0.239	0.301
MAPE	0.542	0.301	0.297	0.380	0.506

## Data Availability

Data is available at https://www.webpages.uidaho.edu/ui-prmd/ (accessed on 7 October 2024) and https://vrai.dii.univpm.it/content/kimore-dataset (accessed on 29 December 2025). The source code of our model is available at https://github.com/BryceLoski21/D2STA (accessed on 29 December 2025).

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
