# Peer review of "Dual-Stream STGCN with Motion-Aware Grouping for Rehabilitation Action Quality Assessment"

_sensors, 2026, doi:10.3390/s26010287_

Round 1
Reviewer 1 Report
Comments and Suggestions for Authors
This paper proposes a rehabilitation motion quality assessment method based on dual-stream spatio-temporal graph convolutional networks and dynamic motion-aware grouping. Its core contribution lies in integrating position and orientation dual-modal features while introducing a dynamic joint grouping mechanism based on motion amplitude. The proposed approach achieves superior performance compared to existing methods on datasets such as KIMORE.
- Although the introduction mentioned existing methods' limitations in modeling joint synergistic relationships, insufficient flexibility in fixed grouping, and reliance on single features, the discussion remains rather general and lacks an in-depth explanation of why these limitations are particularly critical in rehabilitation assessment.
- Lines 33-34: “Existing studies have improved evaluation accuracy through multimodal fusion, spatio-temporal attention mechanisms [6], and graph convolutional networks”, to provide more effective evidence, the authors may consider referring to the following updated relevant studies: Data-Driven Deep Learning for Predicting Ligament Fatigue Failure Risk Mechanisms (https://doi.org/10.1016/j.ijmecsci.2025.110519). This study combines multimodal fusion with advanced attention mechanisms, providing valuable insights for advancing the significance of this research.
- The introduction directly lists the contributions of this paper after identifying the problem, but lacks a step-by-step reasoning on how the proposed method solves the aforementioned issues.
- Formula (6) uses only the standard deviation of joint coordinates over the time dimension as a measure of movement amplitude. This definition is overly simplistic and fails to account for: physiological differences in the range of motion across joints; the fact that the importance of joint movement in rehabilitation exercises is not solely determined by absolute displacement (small-amplitude but high-precision control movements may be more critical); and the metric's strong dependence on the coordinate system, lacking necessary normalization (relative to the torso or root node).
- Although a grouping strategy based on motion amplitude was proposed, it did not specify how the number of groups (G=6) was determined, nor did it analyze the impact of different group sizes on performance.
- In the UI-PRMD dataset, the use of orientation data without positional data is inconsistent with the dual-stream design on KIMORE, and the rationale for this choice and its impact on the comparability of results are not adequately explained.
- No ablation study was conducted on the dynamic grouping module itself (compared to fixed grouping), nor were the effects of varying the number of attention heads or different methods for generating grouping masks tested.
- Only numerical results are presented, lacking analysis of error cases, discussion of the model's strengths and limitations across different action types, and exploration of why it performs slightly worse on certain exercises.
- The study fails to specify the sample distribution and scoring criteria differences between the KIMORE and UI-PRMD datasets, nor does it address data imbalance or cross-dataset generalization capabilities.
- Experimental setup and reproducibility information are insufficient, lacking details on training duration, convergence curves, and hyperparameter tuning processes. It remains unclear whether the open-source code includes complete training and evaluation scripts.
Author Response
We sincerely thank you for your insightful comment. Please see the attachment.

Reviewer 2 Report
Comments and Suggestions for Authors
The manuscript addresses an interesting topic; however, the following clarifications are required:
-
Please specify which exercises are considered in the study. Who performs the exercises, and how many repetitions/performances were carried out for each exercise?
-
Please clarify the nature of the input data. Is the input video data or coordinate data? If coordinates and orientation are used (as shown in Figure 1), describe in detail how these coordinates and orientations are obtained or extracted.
-
It is not clear to the reviewer why the authors perform grouping based on range of motion (ROM). What motivates the need to create such groups? Please provide a thorough explanation and justification for this choice.
-
Replace Figure 2 with a table containing the corresponding numerical values to improve clarity and reproducibility.
-
How is the ground truth defined and calculated? On what basis do the authors determine that an exercise execution is correct?
Please address each point explicitly in the revised manuscript.
Author Response

(The authors gave the same response as above.)

Reviewer 3 Report
Comments and Suggestions for Authors
The abstract doesn't provide the necessary context for this work. Why action quality assessment is useful? What does it do? What is these groupings that the abstract is discussing? The text goes straight to explain what is the problem with current practice. I am aware that these things are explained later in the text, but some context must be provided to the first-time reader
the acronym STGCN (spatio-temporal graph convolutional network) is stated anywhere in the text (the abbreviation (ST-GCN is used) while AQA(Action quality assessment) is explained twice (lines 21 & 74). It is a good idea to explain the acronyms once when they are first mentioned in the text and to be consistent with the abbreviations (either STGCN or ST-GCN). It may not be necessary to state well-known acronyms, such as GCN, but it makes the text more complete.
Introduction,
I suggest move the parts that discuss AQA in the beginning and the parts that discuss GCN and other methods after that. They way it is written now, it feels like each paragraph is disjoint from the previous one.
Related Work:
A few more details regarding the specifics of AQA are needed. Specifically, what metrics are used for AQA, how performance is rated, and why superior classification is required/why current practice is limited. All these components need to be stressed, otherwise the presented work appears to be just another classification method with minimal practical impact.
Method:
The model of the human body needs to be presented. How many joints, their local coordinate frame (there is mention of quaternions in the text), if it is based on an existing model in the literature, total degrees of freedom ideally a figure listing the joints will also be helpful. Lter you mention the dataset that have different models each. How was that handled? Did the motions were mapped on a default model to enable cross-model comparisons or this work is model agnostic, as long as all data are consistent? I can assume that it is the latter, but it is better to state it explicitly.
Line 143: This equation should be presented as equation (1).
Line 152: "..the Loss function L (eq. (2)).." or something similar. The point is to refer to the equation that is presented in the text without confusing the reader. Though the equation is three lines below and easily seen, it is better to be explicit.
line 157: is â„“(Å·, y) = ∥ŷ − y∥ the same as â„“( f (xi ; θ ), yi )?
Line 261: This is not the correct way to describe the motion capture devices. The number of cameras should be mentioned for the vicon, as well as the version of the software and model that was used (eg. CGM version). Similarly, more details for the kinect are needed (kinect 1, kinect 2, Azure kinect, which sdk etc.). Additionally, the capture frequency should be mentioned and if the two systems were synchronised. I am aware that this is how the paper presenting the UI-PRMD describe the methods and it is not negligence from the authors of this work, but the lack of information remains. I recommend stating, in a very respectful and discreet manner, that these are the only information being provided and focus on the dataset.
How did the authors handled the differences in the number of joints between all these models? Where the data completely separated? Was training happened separately for each dataset as well?
line 284: The rehabilitation exercises should be mentioned in the Methods section. It might be important to understand if they were focusing in a specific area (e.g. lower body) and if the performance is impacted by it. For example ex4 in Table 2 is worse than Kuang et al and Yao et al., that might have to do with the motions themselves.
Conclusions:
The conclusion segment needs to state clearly the contribution. Mention that the results on the KIMORE dataset showed more promising results than UI-PRMD dataset and, ideally, comment on the discrepancy. Also comment on the practical applications of the work presented and the potential for future studies fellow researchers should pursue (developing better GCNs, clinical validation of this work etc.).
Author Response

(The authors gave the same response as above.)

Round 2
Reviewer 1 Report
Comments and Suggestions for Authors
All comments have been addressed.
Reviewer 2 Report
Comments and Suggestions for Authors
The authors have addressed the main issues.